# Clinical outcomes after small-incision lenticule extraction versus femtosecond laser-assisted LASIK for high myopia: A meta-analysis

**Yanyan Fu[1,2,3], Yewei Yin[1,2,3☯], Xiaoying Wu[1,2,3☯], Yuanjun Li[1,2,3☯], Aiqun Xiang[1,2,3], Ying Lu[1,2,3], Qiuman Fu[1,2,3], Tu Hu[1,2,3☯], Kaixuan Du[1,2,3], Dan Wen[1,2,3]***

**1** Department of Ophthalmology, XiangYa Hospital, Central South University, Changsha, Hunan, China,
**2** Eye Center of Xiangya Hospital, Central South University, Changsha, Hunan, China, **3** Hunan Key Laboratory of Ophthalmology, Changsha, Hunan, China

☯ These authors contributed equally to this work.
* wendan@csu.edu.cn

## Abstract

### Aim

To compare postoperative clinical outcomes of high myopia after small-incision lenticule extraction (SMILE) and femtosecond laser-assisted laser in situ keratomileusis (FS-LASIK).

### Methods

From March 2018 to July 2020, PubMed, MEDLINE, Embase, the Cochrane Library, and several Chinese databases were comprehensively searched. The studies meeting the criteria were selected and included; the data were extracted by 2 independent authors. The clinical outcome parameters were analyzed with RevMan 5.3.

### Results

This meta-analysis included twelve studies involving 766 patients (1400 eyes: 748 receiving SMILE and 652 receiving FS-LASIK). Pooled results revealed no significant differences in the following outcomes: the logarithm of the mean angle of resolution (logMAR) of postoperative uncorrected distance visual acuity (weighted mean difference (WMD) = -0.01, 95% confidence interval (CI): -0.02 to 0.00, $I^2$ = 0%, P = 0.07 at 1 mo; WMD = -0.00, 95% CI: -0.01 to 0.01, $I^2$ = 0%, P = 0.83 at 3 mo; WMD = -0.00, 95% CI: -0.01 to 0.00, $I^2$ = 32%, P = 0.33 in the long term), and the postoperative mean refractive spherical equivalent (WMD = -0.03, 95% CI: -0.09 to 0.03, $I^2$ = 13%, P = 0.30). However, the SMILE group had significantly better postoperative corrected distance visual acuity (CDVA) than the FS-LASIK group (WMD = -0.04, 95% CI, -0.05 to -0.02, $I^2$ = 0%, P<0.00001). In the long term, postoperative total higher-order aberration (WMD = -0.09, 95% CI: -0.10 to -0.07, $I^2$ = 7%, P<0.00001) and postoperative spherical aberration (WMD = -0.15, 95% CI: -0.19 to -0.11, $I^2$ = 29%, P<0.00001) were lower in the SMILE group than in the FS-LASIK group; a significant difference was also found in postoperative coma (WMD = -0.05, 95% CI: -0.06 to -0.03, $I^2$ = 30%, P<0.00001).

**Data Availability Statement:** All relevant data are within the manuscript and its Supporting information files.

**Funding:** Science and Technology Project of Changsha Project name: Clinical study of small incision lenticule extraction in the treating myopia. Project number: Kq1701079 Changsha Health and Family Planning Commision is responsible for this project.

**Competing interests:** The authors have declared that no competing interests exist.

**Abbreviations:** CDVA, corrected distance visual acuity; FS, Femtosecond laser; FS-LASIK, Femtosecond laser assisted laser in situ keratomileusis; NOS, Newcastle-Ottawa scale; PRISM, Preferred reporting items for systematic reviews and meta-analyses; SE, Spherical equivalent; SMILE, Small incision lenticule extraction; tHOA, Total higher-order aberration; UDVA, Uncorrected distance visual acuity; WMD, Weighted mean difference.

## Conclusion

For patients with high myopia, both SMILE and FS-LASIK are safe, efficacious and predictable. However, the SMILE group demonstrated advantages over the FS-LASIK group in terms of postoperative CDVA, while SMILE induced less aberration than FS-LASIK. It remains to be seen whether SMILE can provide better visual quality than FS-LASIK; further comparative studies focused on high myopia are necessary.

## Introduction

With the increasing prevalence of high myopia, high requirements have been placed on the predictability of refractive surgery and on the visual quality that it achieves [1]. Patients with high myopia face longer and more difficult postoperative wound healing than those with low to moderate myopia; this challenge increases the risk of stromal haze formation and refractive regression and reduces the long-term stability of the refractive correction [2,3]. In addition, high myopia means a high degree of correction during the procedures, and the thin remaining part of the cornea will be at risk for corneal ectasia [4]; thus, it is generally difficult to reach the expected degree of postoperative visual quality [5]. Owing to these conditions, the correction of high myopia carries many challenges for refractive surgeons; failure to use an appropriate correction strategy could lead to significant visual impairment and an elevated risk of sight-threatening complications [6,7]. Consequently, the relative merits of different types of corneal refractive surgery for high myopia are not only a concern for patients but also an important research topic for refractive surgeons.

Recently, small-incision lenticule extraction (SMILE) and femtosecond laser-assisted laser in situ keratomileusis (FS-LASIK) have become the most popular options in corneal refractive surgery. FS-LASIK has proven to be effective, safe and predictable for treating myopia [8]. However, the creation of the corneal flap and the ablation of the stroma limit the application of FS-LASIK, as this procedure may increase the risk of treatment regression, changes in corneal biomechanics, and flap complications. SMILE has emerged as a new option for patients; in this procedure, the production of a corneal flap is replaced by a minimized incision to reduce corneal-flap complications and dry eye [9].

Many scholars have focused on the clinical efficacy of these two types of refractive surgery. However, most studies comparing SMILE and FS-LASIK consider correction of low to moderate myopia [10–13] or analyze all included eyes without further grouping by degree of myopia [14]. Only a few comparative studies have targeted populations with high myopia. Hence, the aim of this meta-analysis is to review on the existing comparative studies in greater depth to understand the differences between SMILE and FS-LASIK in terms of safety, efficacy, predictability, and visual quality when used to correct high myopia.

## Material and methods

A meta-analysis was performed in accordance with the Meta-analysis of Observational Studies in Epidemiology (MOOSE) guidelines [15], following the generally accepted recommendations [16].

### Ethics statement

This study followed the tenets of the Declaration of Helsinki and was approved by the ethics committee of XiangYa Hospital, Central South University. Informed written consent was obtained from all participants.

**Search strategy.** To gather as many records as possible on the comparison between SMILE and LASIK for treating high myopia, two reviewers independently searched the following electronic databases: PubMed, EMBASE, the Cochrane Central Register of Controlled Trials (CENTRAL) and three Chinese databases (CNKI, WANFANG and Weip). The following keywords were used in the search: high myopia (e.g., high myopia, high short-sight, high nearsighted or high correction), LASIK (e.g., LASIK or keratomileusis, femtosecond laser in situ keratomileusis) and SMILE (e.g., SMILE, lenticule extraction, small-incision lenticule extraction). The search process for PubMed is shown in Fig 1 (**Flow diagram of the literature search**).

No date or language restrictions were applied to the electronic search. Our literature search work began in March 2018 and ended in July 2020. During this period, we searched once a month to observe whether there was any newly published literature meeting the inclusion criteria and consider whether to include it. At the end of the last search, we had identified 12 candidate articles. First, two reviewers independently screened the titles and abstracts; then, the potentially relevant reports were assessed to determine whether they were complete manuscripts; finally, the two researchers selected the articles in accordance with our inclusion criteria. Any disagreements between the reviewers were eliminated through discussion, and the two reviewers eventually reached a consensus about the results and interpretation.

**Inclusion criteria.** This meta-analysis adopted the following inclusion criteria for articles: (1) study design: randomized or nonrandomized clinical trials; (2) population: patients with high myopia (preoperative spherical equivalent (SE) refractive error -6.00 diopters or worse, or the simultaneous presence of spherical refractive error worse than -5.00 diopters and cylindrical refractive error worse than -1.00); (3) intervention: SMILE versus FS-LASIK; (4) outcome variables: visual acuity or aberration or other parameters that represent clinical outcomes; (5) data: original clinical articles with independent data; (6) no date restrictions on the included studies.

**Exclusion criteria.** The following classes of articles were excluded: 1) repeated publications; 2) unpublished literature; 3) abstracts, case reports, reviews, letters, comments, noncomparative studies and nonhuman investigations; and 4) reports with incorrect or incomplete data.

**Data extraction.** Two independent reviewers extracted data from the included studies using a customized form. In order to reduce the heterogeneity caused by variation in follow-up intervals, some outcomes are presented in subgroups defined by the follow-up time (e.g., 1 mo, 3 mo or in the long term after surgery). The following parameters were extracted:

1. The primary outcome measures represented postoperative safety, efficacy and predictability, for instance, the logarithm of the mean angle of resolution (logMAR) values of uncorrected distance visual acuity (UDVA), the logMAR values of corrected distance visual acuity (CDVA), and postoperative mean refractive SE.

2. The secondary outcome measures were various objective parameters of aberration suggesting postoperative visual quality, including total higher-order aberration (tHOA), spherical aberration, and coma.

3. If there were multiple reports for a particular study, only the data from the most recent and representative publication were extracted.

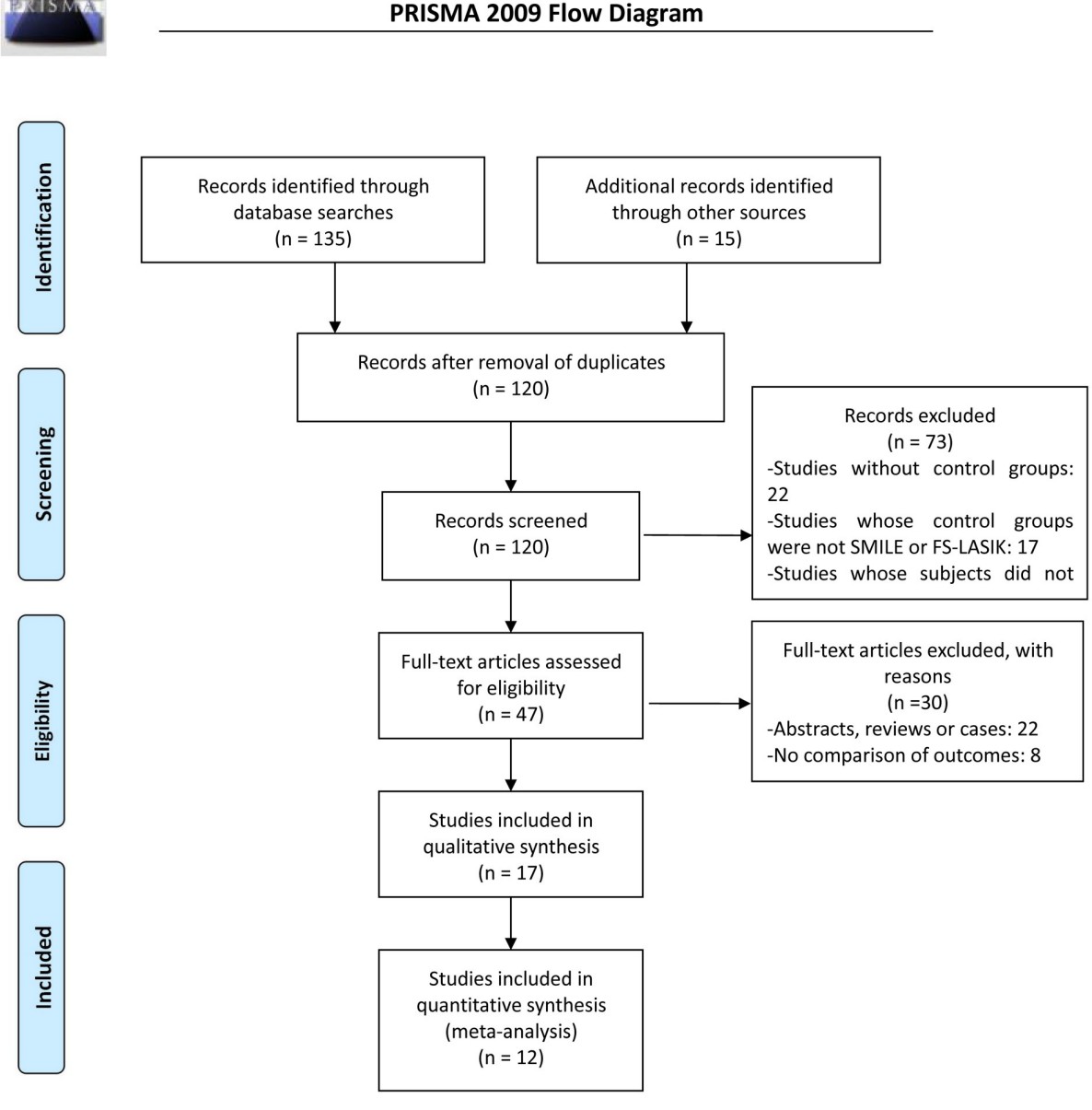

**Fig 1. Flow diagram of the literature search.**

## Quality assessment

Because it is difficult to achieve completely randomized, controlled, and double-blind experimental design in clinical studies of FS-LASIK and SMILE for myopia, only one study [20] was a randomized controlled trial; most of the included studies were nonrandomized comparative trials. The quality of the included studies was assessed with the Newcastle-Ottawa Scale (NOS), which was adopted to evaluate the cohorts. The scores of these 12 included studies are presented in Table 1, along with judgments about each risk-of-bias item for each included study

**Table 1. Characteristics of the 10 included studies.**

| Study or subgroup | Year | Design | Language | SMILE group | | FS-LASIK group | | Follow-up (mo) | NOS |
|---|---|---|---|---|---|---|---|---|---|
| | | | | Eyes (n) | Preop mean SE (D) | Eyes (n) | Preop mean SE (D) | | |
| Bingjie Wang [2] | 2016 | CT | English | 50 | -7.60±1.12 (>-6.00) | 56 | -7.68±1.19 (>-6.00) | 12 mo | 6 |
| Bingjie Wang [3] | 2015 | CT (prospective) | English | 47 | −7.46±1.11 (≥−6.00) | 43 | −7.44±1.13 (≥−6.00) | 12 mo | 7 |
| Yishan Qian [6] | 2020 | CT (prospective) | English | 51 | Sum of the spherical and cylindrical refractive error (-10.00 ~14.00) | 45 | Sum of the spherical and cylindrical refractive error (-10.00 ~14.00) | 6 mo | 7 |
| Tian Han [7] | 2020 | CT | English | 75 | -8.79±1.83 (-6.00 ~12.25) | 46 | -9.17±2.03 (-6.25 ~12.25) | 24 mo | 6 |
| Likun Xia [17] | 2018 | CT (prospective) | English | 78 | -8.11±1.09 (-6.00~-12.00) | 65 | -8.05±1.12 (-6.00~-12.00) | 36 mo | 7 |
| Tian Han [18] | 2018 | CT | English | 60 | -6.54±1.69 (≥−6.00) | 41 | -7.15±1.92 (≥−6.00) | 36 mo | 7 |
| Congrong Jing [19] | 2018 | CT (prospective) | Chinese | 134 | -6.00~-10.00 | 106 | -6.00~-10.00 | 3 mo | 8 |
| Guofu Chen [20] | 2017 | RCT | Chinese | 64 | -9.59±0.57 (>-9.00) | 64 | -9.77±0.56 (>-9.00) | 6 mo | 6 |
| Xiaojing Li [21] | 2015 | CT | English | 55 | Spherical: -6.00±1.39 Cylindrical: -0.66±0.70 (≥−6.00) | 51 | Spherical: -6.18±1.61 Cylindrical: -0.83±0.66 (≥−6.00) | 6 mo | 6 |
| Yueming Zhou [22] | 2016 | CT | Chinese | 66 | -7.58±2.14 (-6.125 ~ -9.75) | 66 | -7.62±1.83 (-6.00~-9.875) | 6 mo | 6 |
| Xueyi Zhou [23] | 2019 | CT | English | 39 | −10.79±0.81 (−10.00~−13.00) | 34 | −11.06±0.99 (−10.00~−14.50) | 24 mo | 6 |
| Iben Bach Pedersen [24] | 2014 | CT | English | 29 | −6.00~−10.5 | 35 | −6.00 to −10.5 | 12 mo | 6 |

*CT: nonrandomized comparative trial; CT (prospective): prospective, nonrandomized comparative trial; RCT: randomized controlled trial

*Preop mean SE (D): preoperative mean refractive spherical equivalent in diopters; the value in parentheses indicates the range of spherical equivalent values.

(Fig 2: **Judgments about each risk-of-bias item for each included study**). The average NOS score of these 12 studies was 6.5 on a scale from 0 (lowest quality) to 9 (highest quality).

## Statistical analysis

Meta-analysis was conducted in the statistical program RevMan 5.3, using weighted mean difference (WMD) and the corresponding 95% confidence interval (CI) to calculate the continuous outcomes. First, we used $I^2$ to test the heterogeneity of the included literature, and fixed-effect modeling was carried out when there was no statistical heterogeneity among studies ($P \geq 0.1$, $I^2 < 50\%$). Conversely, random-effect modeling was used for analysis when the included literature bore significant evidence of statistical heterogeneity ($P < 0.1$, $I^2 > 50\%$). The results are presented as Z values, each corresponding to a P value; P values less than 0.05 were taken to indicate significant differences.

## Sensitivity analysis and publication bias

In order to evaluate the robustness of the statistical model, a sensitivity analysis was carried out by "leave-one-out" analysis, in which we removed each included study in turn and quantified the influence of the individual studies on the pooled estimates. The results showed that when the study by Li-kun Xia et al. was excluded [17], the $I^2$ value of UDVA within the 1 mo and 3 mo subgroups reduced sharply, and the P value showed a stable significant difference.

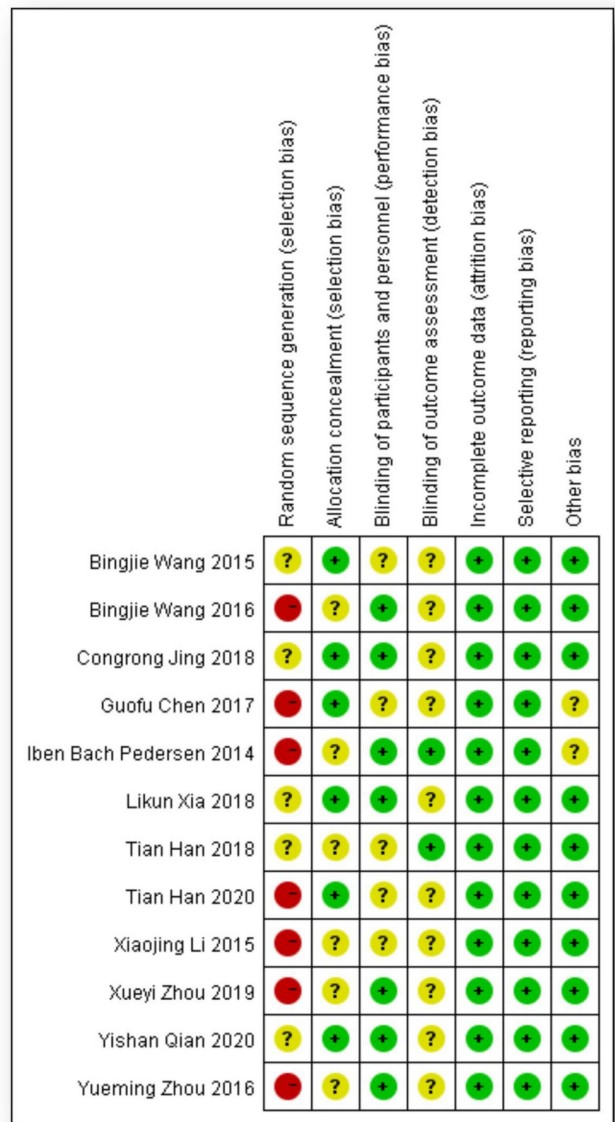

**Fig 2. Judgments about each risk-of-bias item for each included study.**

Publication bias was estimated by applying Egger's test [25] (P = 0.207 to 1.000) and Begg's test [26] (P = 0.246 to 1.000) to the 12 studies; these tests indicated no obvious publication bias.

# Results

## Search results

Fig 1 **(Flow diagram of the literature search)** is a flowchart of the selection of publications in this study. Initially, a total of 150 potentially eligible publications were selected from the electronic databases. After 30 duplicate reports were eliminated, the remaining 120 papers underwent title and abstract screening. Seventy-three studies were excluded for the following

reasons: 22 studies did not have a control group and merely gave separate descriptions of SMILE and FS-LASIK; 17 studies' control groups were not SMILE or FS-LASIK; 34 studies used subjects who did not have high myopia. Ultimately, 12 studies [2,3,6,7,17–24] met our inclusion criteria and were included in this meta-analysis.

## Study characteristics and quality

Table 1 summarizes the main characteristics and the quality assessments of these 12 included studies, which were published from 2014 to 2020. A total of 766 patients (1400 eyes) were evaluated, with 748 eyes in the SMILE group (53%) and 652 eyes in the FS-LASIK group (47%). This meta-analysis identified 1 randomized controlled trial (RCT), 4 prospective comparative studies and 7 nonrandomized comparative studies investigating the effects of SMILE and FS-LASIK in the correction of high myopia. These studies were assessed using the NOS (Table 1), and we also formed judgments about each risk-of-bias item for each included study (Fig 2). Overall, these included studies had good quality (average NOS score: 6.5). S1 Table summarizes the surgical procedures used in the 12 included studies. The laser processes involved in SMILE and FS-LASIK were all performed using the VisuMax femtosecond laser system (Carl Zeiss Meditec), and the cap and flap thicknesses used in the 12 articles are also mentioned in S1 Table.

## Primary outcomes

**The logMAR values of postoperative UDVA.** Of the 12 included articles, 6 [6,7,17–20] reported the logMAR values of postoperative UDVA. We excluded the study by Likun Xia et al. [17] in the first and second subgroups because the overall results were highly sensitive to its outcome. An examination of the forest plot showed that, for high myopia, there was no significant UDVA difference between the SMILE group and the FS-LASIK group at the 1- or 3-month follow-up (WMD = -0.01; 95% CI:-0.02 to0.00; $I^2$ = 0%; P = 0.07, WMD = -0.00; 95% CI:-0.01 to 0.01; $I^2$ = 0%; P = 0.83). These 6 studies were followed up for a longer period of time, and the same results were obtained in the long term after surgery (WMD = -0.00; 95% CI: -0.01 to 0.00, $I^2$ = 32%; P = 0.33). The same was true of the combined results (WMD = -0.00, 95% CI, -0.01 to 0.00, $I^2$ = 0%, P = 0.13; Fig 3A: **Primary outcomes**).

**The logMAR values of postoperative CDVA.** Five studies [6,7,18,19,21] reported the log-MAR values of postoperative CDVA in patients with high myopia. The forest plot indicated that the SMILE group had significantly better postoperative CDVA than the FS-LASIK group in the correction of high myopia (WMD = -0.04, 95% CI: -0.05 to -0.02, $I^2$ = 0%, P<0.00001; Fig 3B: **Primary outcomes**).

**Postoperative mean refractive SE.** Six studies [6,7,17–19,21] compared the postoperative mean refractive SE outcomes between the SMILE and FS-LASIK groups. The forest plot showed no significant difference in postoperative mean refractive SE between the SMILE group and FS-LASIK group (WMD = -0.03, 95% CI: -0.09 to 0.03, $I^2$ = 13%, P = 0.30; Fig 3C: **Primary outcomes**).

## Secondary outcomes

**Aberration.** Four studies [17,18,20,21] presented data on postoperative aberration at long-term follow-ups. We extracted 3-year postoperative data from the studies by Likun Xia [17] and Tian Han [18], and we extracted 6-month postoperative data from the studies by Guofu Chen [20] and Xiaojing Li [21]. Due to measurement bias, there was heterogeneity among these 4 studies; Table 2 shows the differences in the measurement of aberrations in the 4 included studies.

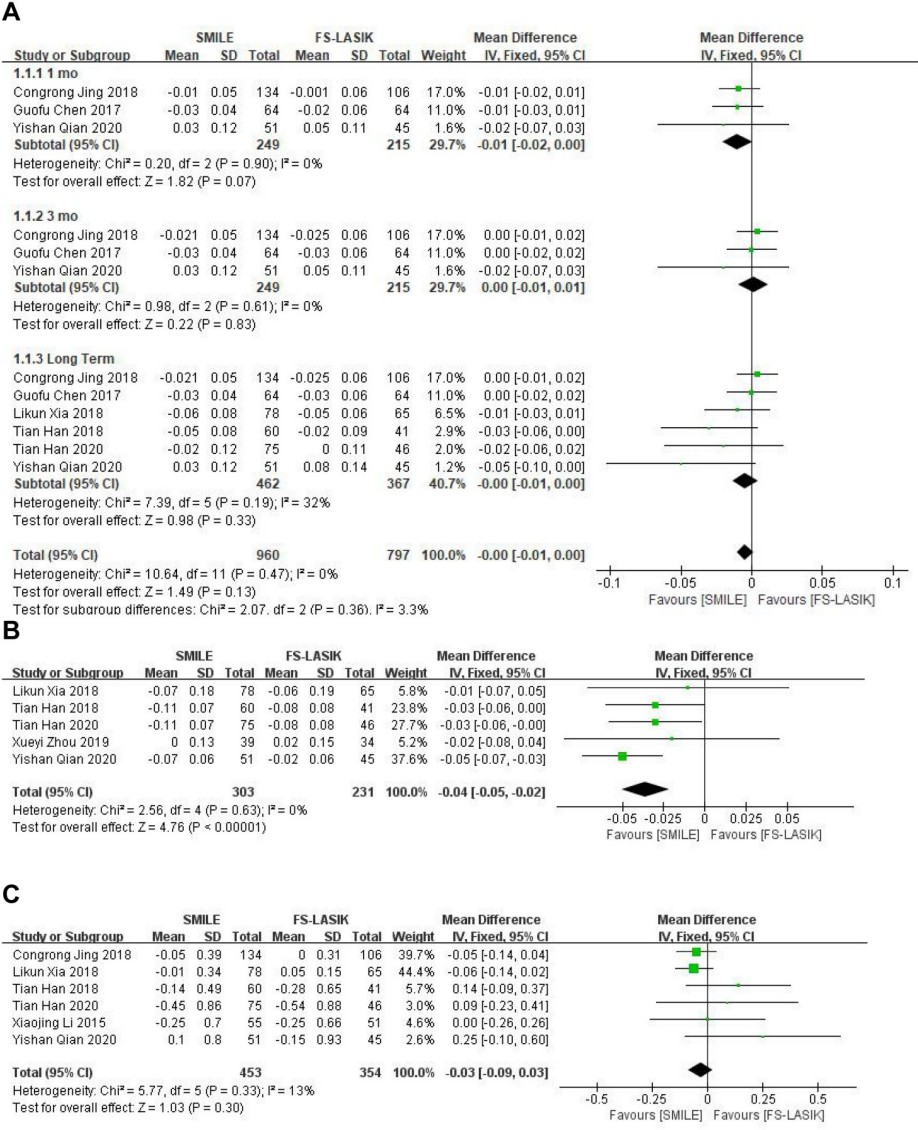

✓ 3A: Forest plot showing the weighted mean difference of UDVA (logMAR) after SMILE versus FS-LASIK
✓ 3B: Forest plot showing the weighted mean difference of CDVA (logMAR) after SMILE versus FS-LASIK
✓ 3C: Forest plot showing the weighted mean difference of postoperative mean refractive SE after SMILE versus FS-LASIK
✓ SD = standard deviation; CI = confidence interval; df = degree(s) of freedom; I² = heterogeneity; Z = overall effect

**Fig 3. Primary outcomes (A-C).**

**Postoperative tHOA.** The tHOA forest plots indicated significant differences between the two groups. For high myopia, the tHOA was increased in both the SMILE group and the FS-LASIK group, but the postoperative tHOA in the SMILE group was significantly lower than that in the FS-LASIK group as of long-term follow-up (WMD = -0.09, 95% CI:-0.10 to -0.07, I² = 7%, P<0.00001; Fig 4A: **Secondary outcomes**).

**Postoperative spherical aberration.** SMILE also introduced less spherical aberration than FS-LASIK as of long-term follow-up (MD = -0.15, 95% CI: -0.19 to -0.11, I² = 29%, P<0.00001; Fig 4B: **Secondary outcomes**).

**Table 2. Measurement bias of 3 included studies.**

| Included studies | Types of refractive surgery | Different measurements of aberrations |
|---|---|---|
| Likun Xia et al [17] | SMILE VS Wavefront-guided FS-LASIK | HOAs, WASCA wavefront analyzer; Carl Zeiss Meditec AG, Jena, Germany |
| Tian Han et al [18] | SMILE VS FS-LASIK | Pentacam HR, Type 70900, Wetzlar, Germany |
| Guofu Chen et al [20] | SMILE VS FS-LASIK | Pentacam; Oculus GmbH, Wetzlar, Germany |
| Xiaojing Li et al [21] | SMILE VS FS-LASIK | Pentacam; Oculus GmbH, Wetzlar, Germany |

**Postoperative coma.** For high myopia, no significant difference in postoperative coma was found between the SMILE group and the FS-LASIK group with long-term postoperative observation (WMD = -0.05, 95% CI: -0.06 to -0.03, $I^2$ = 30%, P<0.00001; Fig 4C: **Secondary outcomes**).

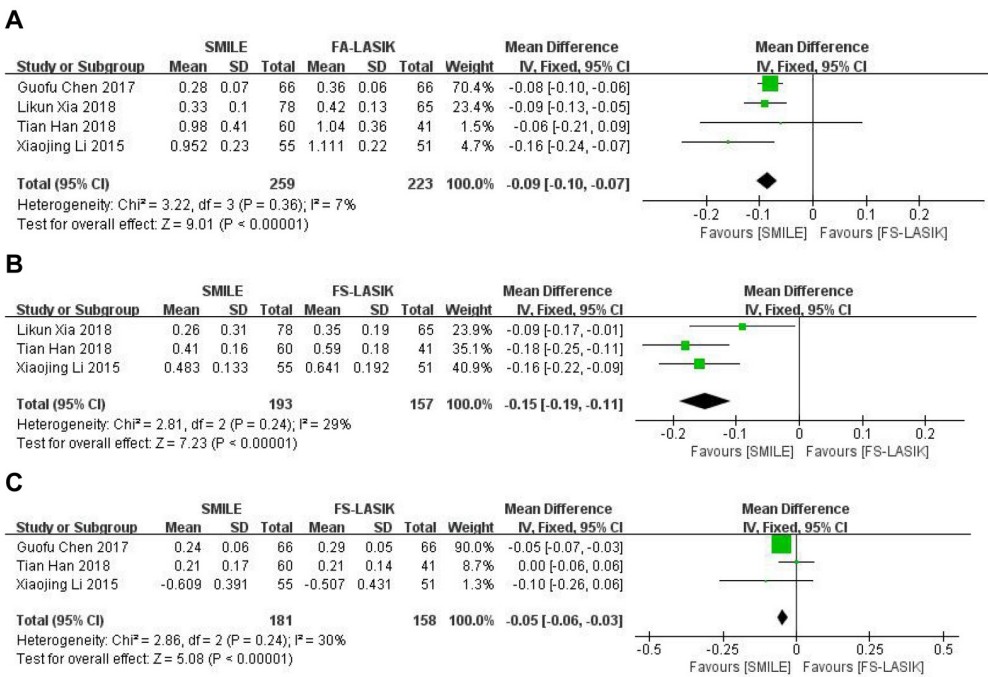

✓  4A：Forest plot showing the weighted mean difference of tHOA after SMILE versus FS-LASIK
✓  4B：Forest plot showing the weighted mean difference of spherical aberration after SMILE versus FS-LASIK
✓  4C: Forest plot showing the weighted mean difference of coma after SMILE versus FS-LASIK

SD = standard deviation; CI = confidence interval; df = degree(s) of freedom; $I^2$ = extent of heterogeneity; Z = overall effect

**Fig 4. Secondary outcomes (A-C).**

## Discussion

This meta-analysis focused on patients with high myopia from the perspectives of postoperative safety, efficacy, predictability and visual quality after SMILE or FS-LASIK and performed a systematic comparative analysis.

During our screening of included studies, we found that published comparative studies of high myopia accounted for only a small proportion of studies comparing SMILE and FS-LASIK (17/104 = 16%). Among these comparative studies, RCTs were rare. It is generally recognized that the results of RCTs are more reliable than those of other experimental designs; we also found that the results of the RCT was always consistent with the combined results, and a similar phenomenon was found in other meta-analyses [27,28]. Thus, it is feasible and important to summarize and compare all published data because doing so can provide refractive surgeons with improved surgical treatment strategies for patients with high myopia. Although there was only 1 RCT out of our 12 included studies, most of the included studies reported long-term follow-up outcomes: 11 studies covered follow-up periods of at least 6 months, 7 studies covered at least 1 year, 4 studies covered at least 2 years, and 3 studies covered 3 years (Table 1). In this meta-analysis, UDVA data were divided into subgroups according to the follow-up time (1 mo, 3 mo, or long term), and data on the other outcomes were extracted for long-term follow-up. In addition, no systematic comparison of such outcomes for high myopia has been published to date, which makes this meta-analysis meaningful.

The pooled results revealed that the SMILE group was not significantly different from the FS-LASIK group in the logMAR values of postoperative UDVA. In terms of efficacy, both SMILE and FS-LASIK brought good visual acuity in patients with high myopia; UDVA was significantly improved after operation. Sensitivity analysis revealed that Likun Xia's study had an outsized statistical influence on the analysis for the logMAR UDVA in the 1 mo subgroup and 3 mo subgroup; therefore, we excluded that study from these subgroups. After Likun Xia's study was excluded, there was no evidence of heterogeneity among the 3 remaining studies; therefore, a fixed-effect model was used in this analysis. This exclusion did not alter the result of the previous analysis, which indicates that the combined results were robust and reliable. Differences in the surgical process may be the major source of heterogeneity in Likun Xia's study (Table 2). Meanwhile, heterogeneity may have arisen from the limited number of studies and other external factors.

In terms of predictability, both groups achieved excellent postoperative mean refractive SE in the 12 included studies. We found no significant differences between the SMILE group and FS-LASIK group with regard to postoperative refractive SE. One included study [20] reported the proportion of eyes with postoperative refractions within ±0.50 D of the targets (90.1% in the SMILE group and 76.6% in the FS-LASIK group) at the 6-month follow-up. Additionally, Ganesh demonstrated that the predictability of the SMILE group exceeded that of the FS-LASIK group because the creation of a flap in FS-LASIK exposes the stroma to hydration changes, leading to inaccurate removal of stromal tissue [24,29]. However, our analysis showed no differences in predictability between the two groups. The reason for this discrepancy may be the use of different laser platforms. There are trials reporting that VisuMax achieved fewer complications than IntraLase [30,31], the platform used for FS-LASIK in Ganesh and Gupta's study. All of the studies included in this meta-analysis used VisuMax.

Most contrastive studies with a SMILE group and an FS-LASIK group showed no significant difference in UCVA, CDVA or postoperative mean refractive SE [3,10,18,19,22,24,32], demonstrating that SMILE and FS-LASIK had comparable efficacy, safety, and predictability for treating myopia. However, when we focused on patients with high myopia, the results differed somewhat from our expectations. The SMILE group showed better postoperative CDVA

than the FS-LASIK group, which suggests that SMILE may have a safety advantage over FS-LASIK when used to correct high myopia. The superior CDVA results of SMILE were also reflected in other articles for high myopia [6,7] and were statistically significant, but there was no difference for low or middle myopia [10,19,24,32]. The reason for this difference may be multifaceted. Andri K. Riau's study [32] suggested that, in vivo, the excimer laser used in LASIK released more cytokines and chemokines than are released in SMILE, recruiting more inflammatory cells to the surgical site. In contrast, SMILE, with a small incision size and femtosecond laser treatment, may result in a reduced wound healing response and corneal inflammation, both of which are important influences on visual acuity. These disparities were significant, especially at higher refractive corrections. In addition, it is worth noting that, during the procedure of FS-LASIK, the time required for stroma ablation is longer at higher degrees of myopia, which means that the corneal stroma bed must be exposed to the air and the laser for a longer time. This greatly reduced the compliance of patients during the operation, which may also account for the increased variability in the safety of LASIK for high myopia. However, this problem did not occur in SMILE because the open corneal flap was replaced by a short incision through which the lenticle was extracted. The duration of the SMILE procedure did not substantially change according to the severity of myopia, which also contributed to the safety advantage of SMILE over FS-LASIK for correcting high myopia.

The literature [33] has noted that, in the early stage after refractive surgery, patients are likely to experience glare, halos, or a decline in night vision as well as a variety of other changes in visual quality. There is some correlation between these changes and the increase in ocular aberration. The secondary outcomes suggested that, in both the SMILE group and the FS-LASIK group, the severity of postoperative aberrations significantly increased with long-term observation. Moreover, FS-LASIK introduced a larger tHOA, spherical aberration and coma than SMILE, which is consistent with the results of some of the studies in this meta-analysis [17,18,20,21].

Most importantly, postoperative tHOA increased more in the FS-LASIK group than in the SMILE group as of long-term follow-up. Possible reasons for the results are as follows. First, the two procedures remove corneal tissue in different manners. The increase in aberration after corneal refractive surgery is mainly induced by the corneal flap and stromal bed [5,13]. Because the deflection or displacement of the corneal flap can lead to a sharp increase in aberration, the effect of the corneal flap on tHOA is more obvious in FS-LASIK than in SMILE [11,34]. Especially in high myopia, some patients have relatively thin corneal flaps designed for safety reasons, placing them at an increased risk for corneal flap deflection or displacement. In contrast, there is no corneal flap in the SMILE process, which eliminates the risk of aberration caused by corneal flap positioning. Second, in SMILE, the small size of the incision and the extraction of the lenticule without a lifted flap reduce the disruption of the peripheral nerve and collagen fibers and preserve the structural integrity of the cornea more than FS-LASIK, which could be an important determining factor for higher-order aberration.

Spherical aberration of the cornea is one of the most important factors limiting the optical quality of the retinal image and the spatial resolution capabilities of the visual system. The occurrence of spherical aberration was influenced by biomechanical factors. SMILE maintains a more hermetic environment during the process of ablation, and the spherical features of the entire cornea are better preserved, which may explain why it introduces less spherical aberration than FS-LASIK in high myopia. In addition, during the process of corneal remodeling, a corneal flap increases the risk of a nonspherical change in the cornea, which will also contributes to the increase in spherical aberration [12].

Many scholars [13,21,34] have indicated that the changes in coma after SMILE and FS-LASIK have distinct characteristics. For instance, the small incision and separation of the lenticule

in SMILE make the process of the wound different from that of FS-LASIK, and there is a considerable change in the coma along the direction of the incision (vertical) with only a small effect in the horizontal direction [13]. Other scholars have posited that the increase in vertical coma after SMILE is related to the imbalanced optical changes along the axis [21]. A certain amount of vertical coma may be beneficial to visual quality in high myopia; however, this view needs to be further discussed. In FS-LASIK, the location of the corneal flap hinge may determine the direction of the introduced coma. If the corneal flap is hinged on the nasal side, the coma along the axis of the flap increases significantly [34], whereas if the flap opens vertically, the coma in the vertical direction increases significantly [13]. Many published articles [35,36] have reported that, in moderate and low myopia, no significant difference was found between the SMILE group and the FS-LASIK group. In studies of high myopia [18,20,21], FS-LASIK always caused a more severe coma than SMILE, which is also consistent with our pooled results on coma. During the ablation process in FS-LASIK, high myopia can increase the amount of time that the corneal flap must remain open; thus, the recovery process will introduce a greater difference along the direction of the corneal flap. In SMILE, however, ablation takes the same amount of time regardless of the degree of myopia, which may explain why FS-LASIK introduces more coma than SMILE in high myopia.

The results of this meta-analysis should be interpreted in the context of several important limitations. First, the number of included clinical trials was relatively small, and only one randomized trial was included, which increased the risk of various types of bias. Second, the processed screening results showed that most of the included studies were performed in Asia, which may have caused publication bias. In addition, the extracted aberration data included various measurements from different wave-front analyzers, which increased the methodological bias.

In conclusion, SMILE and FS-LASIK had comparable safety and efficacy when used for correcting high myopia. However, this analysis indicated that the SMILE procedure may have advantages in some respects, especially for high myopia. SMILE introduced less tHOA, spherical aberration and coma than FS-LASIK. Ultimately, further randomized, double-blinded, prospective studies in high myopia over longer follow-up periods will be necessary to provide better evidence for this conclusion. Additionally, such studies would provide useful guidance in choosing between types of refractive surgery for patients with high myopia.

## Supporting information

**S1 Table. Surgical procedures of the 10 included studies.**
(DOCX)

**S2 Table. Summary of findings.**
(DOCX)

**S1 Checklist. PRISMA NMA checklist of items to include when reporting a systematic review involving a network meta-analysis.**
(PDF)

## Author Contributions

**Conceptualization:** Yanyan Fu, Xiaoying Wu, Dan Wen.

**Data curation:** Yanyan Fu, Aiqun Xiang, Ying Lu, Qiuman Fu.

**Formal analysis:** Yanyan Fu.

**Funding acquisition:** Yanyan Fu.

**Investigation:** Yanyan Fu, Yewei Yin, Aiqun Xiang, Tu Hu, Dan Wen.

**Methodology:** Yanyan Fu, Tu Hu.

**Project administration:** Yanyan Fu, Yewei Yin, Dan Wen.

**Resources:** Yanyan Fu, Ying Lu, Tu Hu.

**Software:** Yanyan Fu, Aiqun Xiang, Ying Lu, Qiuman Fu, Tu Hu.

**Supervision:** Yewei Yin, Xiaoying Wu, Yuanjun Li, Aiqun Xiang, Ying Lu, Dan Wen.

**Validation:** Yanyan Fu, Yewei Yin.

**Visualization:** Yanyan Fu, Yewei Yin.

**Writing – original draft:** Yanyan Fu.

**Writing – review & editing:** Yanyan Fu, Yewei Yin, Xiaoying Wu, Yuanjun Li, Tu Hu, Kaixuan Du, Dan Wen.

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
