## [Decision Letter · Decision Letter 0]

20 Jul 2020

PONE-D-20-10509

Clinical outcomes after small incision lenticule extraction versus femtosecond laser-assisted LASIK for high myopia: a Meta-analysis.

PLOS ONE

Dear Dr. Wen,

Thank you for submitting your manuscript to PLOS ONE. After careful consideration, we feel that it has merit but several points have to be addressed before it can be further considered. Therefore, we invite you to submit a revised version of the manuscript that addresses the points raised during the review process.

We look forward to receiving your revised manuscript.

Kind regards,

Yu-Chi Liu, M.D

Academic Editor

PLOS ONE

Journal Requirements:

2. We note your manuscript is currently in landscape as opposed to portrait layout, could you please update the layout to portrait.

3. Please ensure that all search terms and combinations and the complete search strategy have been provided in the Methods section and/or Supplemental information.

4.We suggest you thoroughly copyedit your manuscript for language usage, spelling, and grammar. If you do not know anyone who can help you do this, you may wish to consider employing a professional scientific editing service.  

5. Your ethics statement must appear in the Methods section of your manuscript. If your ethics statement is written in any section besides the Methods, please move it to the Methods section and delete it from any other section. Please also ensure that your ethics statement is included in your manuscript, as the ethics section of your online submission will not be published alongside your manuscript.

Reviewers' comments:

Reviewer's Responses to Questions

**Comments to the Author**

1. Is the manuscript technically sound, and do the data support the conclusions?

Reviewer #1: Yes

Reviewer #2: No

Reviewer #3: Yes

2. Has the statistical analysis been performed appropriately and rigorously? 

Reviewer #1: Yes

Reviewer #2: Yes

Reviewer #3: Yes

3. Have the authors made all data underlying the findings in their manuscript fully available?

Reviewer #1: Yes

Reviewer #2: Yes

Reviewer #3: Yes

4. Is the manuscript presented in an intelligible fashion and written in standard English?

Reviewer #1: No

Reviewer #2: No

Reviewer #3: No

5. Review Comments to the Author

Reviewer #1: This manuscript summarizes clinical outcomes after small incision lenticule extraction versus femtosecond laser assisted LASIK for high myopia using meta-analysis. I have below questions and comments.

Table 1. It’s not clearly described what the numbers are under SE(D) are. Which number is SE? What’s SE or D?

The funnel plot (Supplementary Figure 1) with statistical test results are not robust because tests for funnel plot asymmetry should not be used when the number of studies is less than 10.

Table 3, What CH?(P<0.001)? What are those numbers are under SMILE or FS-LASIK?

In Figures 3 and 4, please add note/legend to clearly indicate which outcome is summarized under figure A, B, C or D.

Line 171, “All of these 10 included studies were retrospective non-random control trials”. Some of these trials are prospective trials. Please double check.

What are the potential reasons that such trials only performed or reported from Asia?

The font size is too small to review conveniently. Please increase the font size in revisions.

Authors please ask an English editor to edit the writing in English.

Reviewer #2: General comments:

#1 The authors have tried to perform a meta analysis on clinical outcomes of SMILE versus FS-LASIK for high myopia. Such initiatives are welcome, but unfortunately very few well-performed RCTs have been performed. Maybe just one...

#2 I think the authors should focus on one outcome, maybe three clinical outcomes: Safety, efficacy and predictability, and focus their manuscript on these aspects. They should leave out biomechanics changes, aberrations etc. This will make the manuscript more clear (see Specific comments).

#3 The vast majority of included studies are based on Chinese patients. This should be discussed.

#4 The manuscript needs to be revised by a native English person.

Specific comments:

#5 Line 68: Please correct that publications up to November 2019 were included.

#6 Line 223 : Drop "Biomechanical effects"

#7 Line 249 ff: Drop discussion on "glare" as this was not primary outcome.

#8 Line 254: Drop discussion on trauma.

#9 Line 259: Drop discussion on "wound healing"

#10 Line 266: Drop discussion on posterior cornea, as this was not investigated.

#11 Line 277: Drop discussion on contest sensitivity, as this was not investigated.

#12 Line 283: Drop discussion on biomechanics changes.

#13 Line 287: Drop discussion on PCE, as this was investigated.

Reviewer #3: This is an important Meta Analysis, and one that has been carried out carefully by the authors. The only question i have is how the authors have left out one study in one context (for looking at UDVA) and used it in another context (HOA's).

6. PLOS authors have the option to publish the peer review history of their article (what does this mean?). If published, this will include your full peer review and any attached files.

Reviewer #1: No

Reviewer #2: No

Reviewer #3: **Yes: **Rupal Shah

---

## [Author Response · Author response to Decision Letter 0]

22 Aug 2020

We have revised the manuscript according to editors’ comments.

1． Some format has been modified according to PLOS ONE's style requirements.

2． Since the table1 page layout, except table1 other pages layout have been modified to portrait.

3． All search terms and combinations and the complete search strategy have been provided in the revised version Methods section.

4． We have employed a professional scientific editing service (AJE) according to your recommendation to modify the language editing, translation, manuscript formatting, and figure formatting, the “AJE Editing Certificate” was also uploaded in the Supplemental information.

5． Our ethics statement have been added in the Methods section of revised version manuscript.

6． Because of the time interval, we have re-searched all the databases and add two included studies to this meta- analysis according to the screening process, Except for CDVA, there was no significant change in other results.

Reviewer #1: This manuscript summarizes clinical outcomes after small incision lenticule extraction versus femtosecond laser assisted LASIK for high myopia using meta-analysis. I have below questions and comments.

Table 1. It’s not clearly described what the numbers are under SE(D) are. Which number is SE? What’s SE or D?

Answer: SE means postoperative mean refractive spherical equivalent, and D means diopter. All values in the “SE(D)” column represent the postoperative mean refractive spherical equivalent in diopters. We have revised “SE(D)” to “Preop mean SE (D)” and added comments below Table 1.

The funnel plot (Supplementary Figure 1) with statistical test results are not robust because tests for funnel plot asymmetry should not be used when the number of studies is less than 10.

Answer: We have removed Supplementary Figure 1.

Table 3, What CH?(P<0.001)? What are those numbers are under SMILE or FS-LASIK?

Answer: To make this manuscript clearer, we have left out biomechanical parameters including posterior corneal elevation (PCE) changes as well as CH and CRF values.

In Figures 3 and 4, please add note/legend to clearly indicate which outcome is summarized under figure A, B, C or D.

Answer: We have added notes under Figures 3 and 4.

Line 171, “All of these 10 included studies were retrospective non-random control trials”. Some of these trials are prospective trials. Please double check.

Answer: We have checked and modified it (lines 242-243).

What are the potential reasons that such trials only performed or reported from Asia?

Answer: We have added the following explanation (lines 414-416): “… the processed screening results showed that most of the included studies were performed in China, which may have caused publication bias.”

The font size is too small to review conveniently. Please increase the font size in revisions.

Answer: We have standardized the font size.

Authors please ask an English editor to edit the writing in English.

Answer: We have invited a scientific editor at American Journal Experts (AJE) to edit the English in our manuscript.

Reviewer #2: General comments:

#1 The authors have tried to perform a meta analysis on clinical outcomes of SMILE versus FS-LASIK for high myopia. Such initiatives are welcome, but unfortunately very few well-performed RCTs have been performed. Maybe just one...

Answer: We have added the following explanation (lines 413-414): “… only one randomized trial was included, which increased the risk of various types of bias.”

#2 I think the authors should focus on one outcome, maybe three clinical outcomes: Safety, efficacy and predictability, and focus their manuscript on these aspects. They should leave out biomechanics changes, aberrations etc. This will make the manuscript more clear (see Specific comments).

Answer: To make this manuscript clearer, we have left out biomechanical parameters including posterior corneal elevation (PCE) changes as well as CH and CRF values.

#3 The vast majority of included studies are based on Chinese patients. This should be discussed.

Answer: We have added the following explanation: “… the processed screening results showed that most of the included studies were performed in China, which may have caused publication bias.”

#4 The manuscript needs to be revised by a native English person.

Answer: We have invited a scientific editor at American Journal Experts (AJE) to edit the English in our manuscript.

Specific comments:

#5 Line 68: Please correct that publications up to November 2019 were included.√

PS: We performed the literature searches from March 2018 to July 2020; we did not mean to imply that the latest publication was from July 2020.

#6 Line 223 : Drop "Biomechanical effects"√

#7 Line 249 ff: Drop discussion on "glare" as this was not primary outcome. √

#8 Line 254: Drop discussion on trauma. √

#9 Line 259: Drop discussion on "wound healing"√

#10 Line 266: Drop discussion on posterior cornea, as this was not investigated. √

#11 Line 277: Drop discussion on contest sensitivity, as this was not investigated. √

#12 Line 283: Drop discussion on biomechanics changes. √

#13 Line 287: Drop discussion on PCE, as this was investigated. √

Reviewer #3: This is an important Meta Analysis, and one that has been carried out carefully by the authors. The only question i have is how the authors have left out one study in one context (for looking at UDVA) and used it in another context (HOA's).

Answer: That study (Likun Xia et al. [11]) showed an outsized influence on the results in the “leave-one-out” analysis for UDVA (I2=69% when included, but I2=0% when excluded). However, the “leave-one-out” analyses for tHOA, spherical aberration and coma did not show that those outcomes were highly sensitive to Xia et al. In addition, the results of the statistical analysis were the same whether that study was included or excluded.

---

## [Decision Letter · Decision Letter 1]

24 Sep 2020

PONE-D-20-10509R1

Clinical outcomes after small incision lenticule extraction versus femtosecond laser-assisted LASIK for high myopia: a meta-analysis

PLOS ONE

Dear Dr. Wen,

Thank you for submitting your manuscript to PLOS ONE. We invite you to submit a revised version of the manuscript that addresses the points raised during the review process.

We look forward to receiving your revised manuscript.

Kind regards,

Yu-Chi Liu, M.D

Academic Editor

PLOS ONE

Reviewers' comments:

Reviewer's Responses to Questions

**Comments to the Author**

1. If the authors have adequately addressed your comments raised in a previous round of review and you feel that this manuscript is now acceptable for publication, you may indicate that here to bypass the “Comments to the Author” section, enter your conflict of interest statement in the “Confidential to Editor” section, and submit your "Accept" recommendation.

Reviewer #1: All comments have been addressed

Reviewer #2: (No Response)

Reviewer #3: All comments have been addressed

2. Is the manuscript technically sound, and do the data support the conclusions?

Reviewer #1: (No Response)

Reviewer #2: Yes

Reviewer #3: Yes

3. Has the statistical analysis been performed appropriately and rigorously? 

Reviewer #1: (No Response)

Reviewer #2: Yes

Reviewer #3: Yes

4. Have the authors made all data underlying the findings in their manuscript fully available?

Reviewer #1: (No Response)

Reviewer #2: Yes

Reviewer #3: Yes

5. Is the manuscript presented in an intelligible fashion and written in standard English?

Reviewer #1: (No Response)

Reviewer #2: Yes

Reviewer #3: Yes

6. Review Comments to the Author

Reviewer #1: (No Response)

Reviewer #2: My comments have been addressed, but the abstract have an error which needs to be corrected: In the Results section, the authors write "Pooled results revealed no significant differences ..., the logMAR of postoperative corrected distance visual acuity (CDVA; WMD=-0.04, 95% CI, -0.05 to 0.02, I2=0%, P<0.00001), ..." Think the correct 95% CI is -0.05 to -0.02, which the P value also indicate, and this is also written in the Results section in the manuscript.

Thus, the sentence needs to be corrected.

Reviewer #3: (No Response)

7. PLOS authors have the option to publish the peer review history of their article (what does this mean?). If published, this will include your full peer review and any attached files.

Reviewer #1: No

Reviewer #2: No

Reviewer #3: No

---

## [Author Response · Author response to Decision Letter 1]

28 Sep 2020

Reviewer #1: (No Response)

Reviewer #2: My comments have been addressed, but the abstract have an error which needs to be corrected: In the Results section, the authors write "Pooled results revealed no significant differences ..., the logMAR of postoperative corrected distance visual acuity (CDVA; WMD=-0.04, 95% CI, -0.05 to 0.02, I2=0%, P<0.00001), ..." Think the correct 95% CI is -0.05 to -0.02, which the P value also indicate, and this is also written in the Results section in the manuscript.

Thus, the sentence needs to be corrected.

Answer: We have corrected the Results section in abstract. (line88-93)

Reviewer #3: (No Response)

---

## [Editor Report · Decision Letter 2]

30 Sep 2020

PONE-D-20-10509R2

Clinical outcomes after small incision lenticule extraction versus femtosecond laser-assisted LASIK for high myopia: a meta-analysis

PLOS ONE

Dear Dr. Wen,

Thank you for submitting your manuscript to PLOS ONE. We invite you to submit a revised version of the manuscript that addresses the points raised during the review process.

We look forward to receiving your revised manuscript.

Kind regards,

Yu-Chi Liu, M.D

Academic Editor

PLOS ONE

Additional Editor Comments (if provided):

Please check the accuracy of the statistics the reviewer mentioned. The P value is significant but the CI includes 0.

---

## [Author Response · Author response to Decision Letter 2]

12 Oct 2020

Additional Editor Comments (if provided):

Please check the accuracy of the statistics the reviewer mentioned. The P value is significant but the CI includes 0.

Answer: We have modified all the place that CI values was incorrect.

---

## [Editor Report · Decision Letter 3]

27 Oct 2020

Clinical outcomes after small-incision lenticule extraction versus femtosecond laser-assisted LASIK for high myopia: a meta-analysis.

PONE-D-20-10509R3

Dear Dr. Wen,

We’re pleased to inform you that your manuscript has been judged scientifically suitable for publication and will be formally accepted for publication once it meets all outstanding technical requirements.

Kind regards,

Yu-Chi Liu, M.D

Academic Editor

PLOS ONE
---

## [Editor Report · Acceptance letter]

16 Dec 2020

PONE-D-20-10509R3 

Clinical outcomes after small-incision lenticule extraction versus femtosecond laser-assisted LASIK for high myopia: a meta-analysis. 

Dear Dr. Wen:

I'm pleased to inform you that your manuscript has been deemed suitable for publication in PLOS ONE. Congratulations! Your manuscript is now with our production department. 

Kind regards, 

on behalf of

Dr. Yu-Chi Liu 

Academic Editor

PLOS ONE